# Assessment of Photocatalytic Hydrogen Production from Biomass or Wastewaters Depending on the Metal Co-Catalyst and Its Deposition Method on TiO$_2$

**Mikel Imizcoz** and **Alberto V. Puga** *,†

Instituto de Tecnología Química, Universitat Politècnica de València-Consejo Superior de Investigaciones Científicas, Avenida de los Naranjos, s/n, 46022 Valencia, Spain
* Correspondence: alberto.puga@urv.cat; Tel.: +34-977-55-8740
† Current address: Departament d'Enginyeria Química, Universitat Rovira i Virgili, Avinguda dels Països Catalans, 26, 43007 Tarragona, Spain

**Abstract:** A systematic study on the solar photocatalytic hydrogen production (photoreforming) performance of $M$/TiO$_2$ ($M$ = Au, Ag, Cu or Pt) using glucose as a model substrate, and further extended to lignocellulose hydrolysates and wastewaters, is herein presented. Three metal ($M$) co-catalyst loading methods were tested. Variation of the type of metal results in significantly dissimilar H$_2$ production rates, albeit the loading method exerts an even greater effect in most cases. Deposition-precipitation (followed by hydrogenation) or photodeposition provided better results than classical impregnation (followed by calcination). Interestingly, copper as a co-catalyst performed satisfactorily as compared to Au, and slightly below Pt, thus representing a realistic inexpensive alternative to noble metals. Hydrolysates of either α-cellulose or rice husks, obtained under mild conditions (short thermal cycles at 160 °C), were rich in saccharides and thus suitable as feedstocks. Nonetheless, the presence of inhibiting byproducts hindered H$_2$ production. A novel photocatalytic UV pre-treatment method was successful to initially remove the most recalcitrant portion of these minor products along with H$_2$ production (17 μmol g$_{cat}$$^{-1}$ h$^{-1}$ on Cu/TiO$_2$). After a short UV step, simulated sunlight photoreforming was orders of magnitude more efficient than without the pre-treatment. Hydrogen production was also directly tested on two different wastewater streams, that is, a municipal influent and samples from operations in a fruit juice producing plant, with remarkable results obtained for the latter (up to 115 μmol g$_{cat}$$^{-1}$ h$^{-1}$ using Au/TiO$_2$).

**Keywords:** photocatalysis; hydrogen; biomass; wastewaters; cellulose; rice husks

## 1. Introduction

Biomass represents a sustainable alternative to fossil resources as feedstock for the production of H$_2$ [1,2]. The importance of hydrogen in fuel cells for clean energy generation can be coupled in this manner with the utilization of renewable feedstocks by existing, well-known, thermochemical routes. Unfortunately, thermochemical production processes such as gasification are energy intensive, often requiring extremely high temperatures, and usually leading to degradation or side reactions, which have detrimental effects on H$_2$ selectivity [3,4]. Advanced aqueous phase reforming reactions producing mixtures of H$_2$, CO$_2$ and light hydrocarbons on appropriate supported metal catalysts have been reported for biomass derivatives, yet selectivities to hydrogen remain moderate [5,6]. The biomass reforming process can be also triggered by light in a photocatalytic process known as photoreforming [7–9]. This process proceeds at ambient temperatures, and thus, the extent of degradation reactions is negligible, resulting in extremely high H$_2$ selectivities. If properly designed,

the photoreforming approach may combine the benefits of reductive $H_2$ production, waste valorization and oxidative biomass conversion into selected chemicals.

Direct photoreforming of lignocelluloses is challenging due to their inertness and insolubility, requiring highly basic media [10–15]. Some efforts have been also dedicated to designing photocatalytic systems able to hydrolyze lignocelluloses and photoreform the resulting saccharides in a cascade fashion [16–20], or to combining mechanochemistry and photocatalysis [21]. Yields and selectivities of these technologies have been optimized noticeably, albeit efficiencies tend to be moderate. Performing the biomass-to-hydrogen transformation in two steps via hydrolysis-photoreforming (as depicted in Figure 1a) might allow a better control over each process, although the few investigations based on this approach reveal issues such as the generation of recalcitrant intermediates inhibiting $H_2$ evolution [22,23].

Photocatalytic valorization is particularly interesting if it is capable of transforming waste into energy or chemicals by using only sunlight as the promoter [24,25]. Applied to biomass photoreforming, it is relevant for agricultural waste materials, which are generally burned for heat value or buried in landfills. On a different perspective, the direct utilization of wastewaters containing organic matter, which may be transformed into $H_2$ by photoreforming, represents a reasonable alternative to other treatment methods such as anaerobic fermentation (see Figure 1b), especially for waters of high toxicity [26,27]. The main drawback of using complex aqueous streams for photocatalytic production of hydrogen lies in the possible presence of substances which may inhibit the reaction by either blocking active sites or consuming photo-generated electrons [28]. In this work, we propose two different integral processes for solar photocatalytic $H_2$ production from renewable resources (see Figure 1). The first one entails two steps, namely the transformation of lignocelluloses ($\alpha$-cellulose or rice husks) by hydrolysis and the photoreforming of the resulting aqueous saccharides into hydrogen. The second one is based on the direct valorization of either municipal or industrial wastewaters for sunlight-induced $H_2$ production.

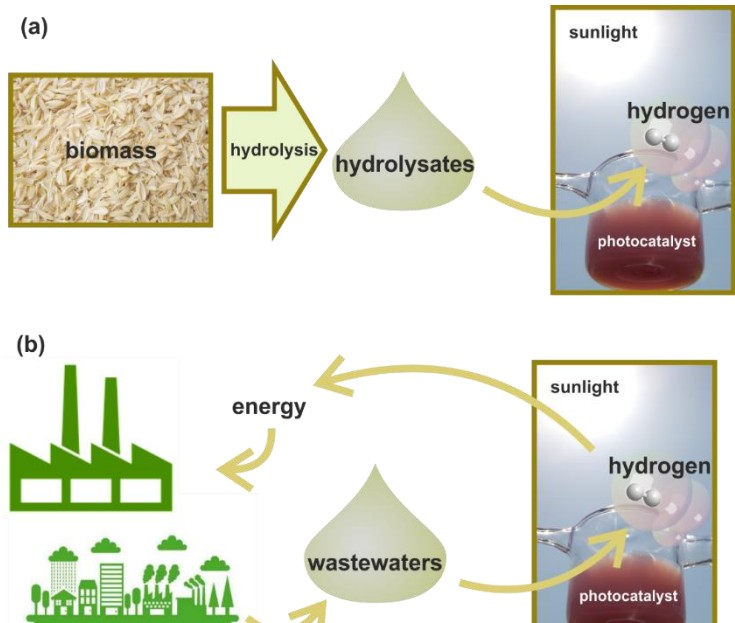

**Figure 1.** Schematic representation of the two processes advocated in this work: (**a**) Two-step biomass-to-hydrogen based on hydrolysis followed by solar photoreforming of the oxygenates contained in the resulting hydrolysates, and (**b**) direct solar photoreforming of either industrial or municipal wastewaters rich in organic matter, whereby the produced $H_2$ can be used for energetic valorization.

## 2. Results and Discussion

### 2.1. Effect of the Metal Co-Catalyst Deposition Method on Loadings and Morphology

The metal co-catalysts in the *M*/TiO$_2$ materials used in this work were deposited by three different methods, namely: (i) Deposition-precipitation, (ii) impregnation and (iii) photodeposition, in order to systematically study the influence of different synthetic protocols on loadings and on co-catalyst nanoparticles morphology, and ultimately, on photocatalytic performance for hydrogen generation from biomass or wastewaters. Experimental composition results are listed in Table 1. Nominal co-catalyst loadings aimed to approach ca. 1% (*w/w* vs. TiO$_2$), yet the actual metallic contents differed from the expected values in some of the studied photocatalysts.

In the case of $^{H,DP}$Au/TiO$_2$, deposition efficiencies are known to remain moderate [29–31], and thus a slight excess of metallic precursor was used (see Section 3.2 below), resulting in an acceptable loading efficiency (≈57%). This method of preparation was selected due to the high proven performance in photocatalytic hydrogen production, as observed in previous investigations in our laboratories [32]. Analogous gold materials prepared by impregnation and photodeposition showed metal loadings closer to their nominal values (see Tables 1 and 2). In the case of $^{IM}$Au/TiO$_2$, metal content was slightly higher than the nominal value, presumably due to a higher than expected gold content of the commercial precursor—a fact that would imply a lower content of water than the three constitutional equivalents. This was confirmed by inductively coupled plasma (ICP) measurements of a stock solution of the gold precursor, revealing a ca. + 10% deviation.

The metal content in the Ag/TiO$_2$ photocatalysts was challenging to determine by ICP due to the formation of precipitates or deposits during dissolution, which led to metal percentages well below the nominal. For this reason the elemental analysis was performed by means of field emission scanning electron microscopy coupled to energy dispersive X-ray spectroscopy (FESEM-EDX), giving values closer to the nominal loadings. Quantitative deposition was observed for $^{PD}$Cu/TiO$_2$, whereby metallic loading was close to the nominal value. The materials prepared by either deposition-precipitation or impregnation had higher copper contents, based on ICP, than the expected value. As for the gold photocatalysts discussed above, this may be attributed to an unknown degree of hydration of the commercial metallic precursor, probably lower than expected. For the Pt/TiO$_2$ photocatalysts the measured metal loading almost matched the nominal value, being slightly lower for the photodeposition than for the deposition-precipitation method (as opposed to gold or silver co-catalysts, see Table 1).

A complete characterization study on the reported *M*/TiO$_2$ materials, including spectroscopic measurements in order to ascertain the chemical characteristics of the co-catalysts (that is, oxidation states and local environments) is underway and will be published elsewhere. Notwithstanding, a preliminary account on metal co-catalyst particle sizes and distributions, as determined from scanning-transmission electron microscopy (STEM) images is herein included (see Table 1) for a tentative guide on morphology-activity relationships.

Regarding the Au/TiO$_2$ materials, the deposition-precipitation method resulted in smaller and more uniformly surface-distributed particles than photodeposition (see Table 1). In addition, the particle size distribution is narrower for $^{H,DP}$Au/TiO$_2$. In the case of $^{IM}$Au/TiO$_2$, gold was not readily found since dispersion was poor, giving rise to a little number of particles, yet of a noticeably large size (in the order of 10$^2$ nm, see Table 1). A similar tendency was observed for Ag/TiO$_2$, that is, particles resulting from the deposition-precipitation method showed the smallest mean diameters, more uniform dispersion, and narrower particle size distribution. In contrast, photodeposition led to a bimodal distribution ranging from small nanometer-scale nanoparticles to a second group larger by an order of magnitude, whereas the growth of Ag crystallites was even more favored by impregnation. Copper domains in Cu/TiO$_2$ were, as expected, significantly more challenging to distinguish from the STEM images, due to the low contrast of such metal with respect to titanium in the support, yet also due to small Cu nanoparticles sizes (and low content) for the three deposition methods studied. Nevertheless, careful elemental analysis measurements allowed the detection of such Cu nanoparticles, as detailed

in our previous work on $^{PD}Cu/TiO_2$ [33]. The particle sizes of Pt nanoparticles were notably smaller than those of the Au and Ag analogues (see Table 1), even for impregnated materials. Photodeposition was, as in the case of silver, slightly different with the occurrence of both ultra-small nanoparticles (around or below 1 nm) and slightly larger ones.

In general, deposition-precipitation gives rise to the smallest particles for all co-catalysts, whereas Au or Ag domains tend to grow significantly by impregnation. Copper and platinum nanoparticles are in all cases smaller than those of gold and silver analogues. Effects of particle sizes with photocatalytic activity will be discussed below.

**Table 1.** Experimental metal (*M*) loadings and particle sizes of *M*/TiO$_2$ (*M* = Au, Ag, Cu or Pt).

| Photocatalyst Code | Loading (*M*,ICP) [a]/ % *w*(*M*)/*w*(TiO$_2$) | Loading (*M*,FESEM-EDX) [b]/ % *w*(*M*)/*w*(TiO$_2$) | Particle Size [c]/nm |
|---|---|---|---|
| $^{H,DP}Au/TiO_2$ | 0.71 ± 0.03 | – [d] | 6.1 ± 2.3 |
| $^{PD}Au/TiO_2$ | 1.06 ± 0.05 [e] | – [d] | 12.4 ± 6.3 |
| $^{IM}Au/TiO_2$ | 1.13 ± 0.04 | – [d] | 104 ± 29 [f] |
| $^{H,DP}Ag/TiO_2$ | – [g] | 0.91 ± 0.32 | 8.8 ± 2.5 |
| $^{PD}Ag/TiO_2$ | – [g] | 1.38 ± 0.35 | 18.7 ± 10.5 [h] |
| $^{IM}Ag/TiO_2$ | – [g] | 1.00 ± 0.22 | 51 ± 40 [f] |
| $^{H,DP}Cu/TiO_2$ | 1.56 ± 0.05 [e] | – [d] | 1.1 ± 0.3 |
| $^{PD}Cu/TiO_2$ | 0.97 ± 0.05 [e] | – [d] | 2.1 ± 1.4 |
| $^{IM}Cu/TiO_2$ | 1.21 ± 0.05 [e] | – [d] | 1.5 ± 0.4 |
| $^{H,DP}Pt/TiO_2$ | 0.98 ± 0.05 [e] | 0.98 ± 0.12 | 0.9 ± 0.3 |
| $^{PD}Pt/TiO_2$ | 0.87 ± 0.03 | 0.84 ± 0.17 | 3.5 ± 3.0 [h] |
| $^{IM}Pt/TiO_2$ | – [g] | 0.91 ± 0.23 | 1.0 ± 0.3 |

[a] Determined by ICP after averaging over a minimum of two measurements. [b] Determined by FESEM-EDX after averaging over several different areas. [c] Determined by measuring the diameters of a number of particles in STEM images. [d] Not determined. [e] Only one measurement done, standard deviation is a conservative estimate based on instrument calibration and data from other similar samples. [f] Large errors are due to limited number of viable particle measurements. [g] Samples could not be dissolved in the acidic analysis solvent. [h] Large errors are due to bimodal particle size distributions.

**Table 2.** Synthetic details for the preparation of *M*/TiO$_2$ (*M* = Au, Ag, Cu or Pt) photocatalysts.

| Photocatalyst Code [a] | Metallic Precursor | Nominal loading (*M*)/% *w*(*M*)/*w*(TiO$_2$) | pH | *t*(Deposition)/h | *v*/mL | Final Color |
|---|---|---|---|---|---|---|
| $^{H,DP}Au/TiO_2$ | HAuCl$_4$·3H$_2$O | 1.25 | 9.7 | 2 | 100 | purple |
| $^{PD}Au/TiO_2$ | HAuCl$_4$·3H$_2$O | 1.15 | – [b] | 3 | 25 | purple |
| $^{IM}Au/TiO_2$ | HAuCl$_4$·3H$_2$O | 1.00 | – [b] | – [c] | ≈1.5 | purple |
| $^{H,DP}Ag/TiO_2$ | AgNO$_3$ | 1.31 | 9.8 | 2 | 54 | purple |
| $^{PD}Ag/TiO_2$ | AgNO$_3$ | 1.08 | – [b] | 3 | 25 | purple |
| $^{IM}Ag/TiO_2$ | AgNO$_3$ | 1.00 | – [b] | – [c] | ≈1.5 | white |
| $^{H,DP}Cu/TiO_2$ | Cu(NO$_3$)$_2$·5/2H$_2$O | 1.24 | 8.5 | 2 | 100 | pale blue |
| $^{PD}Cu/TiO_2$ | Cu(NO$_3$)$_2$·3H$_2$O | 1.00 | – [b] | 3 | 25 | pale blue |
| $^{IM}Cu/TiO_2$ | Cu(NO$_3$)$_2$·5/2H$_2$O | 1.00 | – [b] | – [c] | ≈1.5 | pale turquoise |
| $^{H,DP}Pt/TiO_2$ | H$_2$PtCl$_6$ (aq., 8% *w*/*v*) | 1.00 | 8.7 | 6 | 50 | grey |
| $^{PD}Pt/TiO_2$ | H$_2$Cl$_6$Pt·6H$_2$O | 1.00 | – [b] | 3 | 25 | grey |
| $^{IM}Pt/TiO_2$ | H$_2$Cl$_6$Pt·6H$_2$O | 1.00 | – [b] | – [c] | ≈1.5 | pale ochre |

[a] The superscripted codes at the beginning of the photocatalyst names denote the deposition method as follows: H,DP = deposition-precipitation followed by reduction under hydrogen, PD = photodeposition, and IM = impregnation followed by calcination. [b] Uncorrected pH. [c] Impregnation was performed using the minimum volume of precursor solution to obtain a homogeneous paste after mixing for 1–2 min.

## 2.2. Co-Catalyst Screening

A systematic study on the effect of co-catalyst metal and loading procedure on photocatalytic performance was undertaken using glucose as a model substrate, given that not only does it represent the structural building block and motif of the major part of lignocellulosic biomass (cellulose) waste,

it is also one of the possible organic components in wastewaters generated in the food and drink industrial sectors, as considered in this work (see below).

The hydrogen production results derived from this co-catalyst screening presented in Figure 2 (for a complete set of data, see Table S1) prove that the performance of the studied titania-based photocatalyst is significantly influenced by both the nature of the metal and the loading method. A first glimpse at the data reveal major differences between, on one side, the materials prepared by impregnation and subsequent calcination ($^{IM}M/TiO_2$), and those treated by reductive methods ($^{H,DP}M/TiO_2$ and $^{PD}M/TiO_2$). Impregnation followed by calcination gives rise to materials of moderate activity (below 300 $\mu mol(H_2)$ $g_{cat}^{-1}$ $h^{-1}$). Furthermore, differences across the metal series are not significant for $^{IM}M/TiO_2$. Loading of metallic co-catalysts by photodeposition or deposition-precipitation methods results in significantly higher $H_2$ production rates. For both $^{H,DP}M/TiO_2$ and $^{PD}M/TiO_2$, hydrogen production rates increase along the following order: Ag < Cu < Au < Pt. This is in part coincident with previous photoreforming studies [7], yet most of the reports including metal screening with glucose as the substrate were performed under UV-intense irradiations [34,35], and hence, comparisons should be taken as barely indicative.

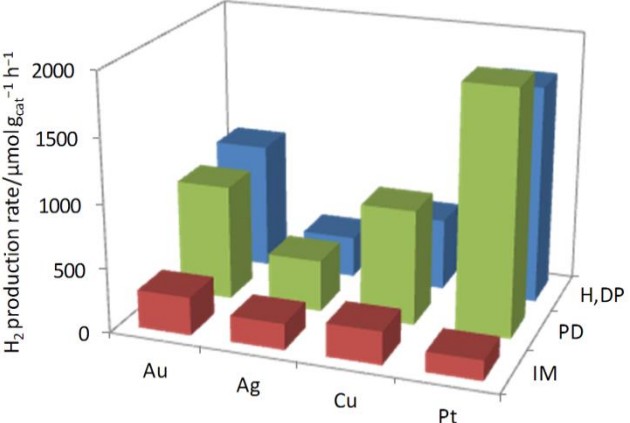

**Figure 2.** Photocatalytic hydrogen production rates from aqueous glucose on $M/TiO_2$ ($M$ = Au, Ag, Cu or Pt). Reaction conditions: Glucose (aq., 5% *w/v*, 25 mL), suspended $M/TiO_2$ (25 mg), simulated sunlight irradiation (AM1.5G, 1.0 kW m$^{-2}$) under Ar atmosphere (1.4 bar) at 25 °C, *t* = 2 h. Estimated standard deviations for $H_2$ production rates lie within a ± 5% error bar, as determined by independent results.

For a more insightful understanding on the intrinsic activity of each co-catalyst, $H_2$ production rates were converted into site-specific—i.e., TOF—values (see Table S1). It is interesting to note that TOFs increase with particle size in all cases. Moreover, reductive loading methods produce more active metal co-catalyst surfaces for Cu and Pt, but the trend is the opposite for Au and Ag, whereby extra-large particles were formed upon impregnation-calcination. Ongoing spectroscopic studies, inspired by relevant prior art dedicated to examining morphology-activity relationships [36–39], are expected to provide more information on other factors influencing photocatalytic activity.

Improved weight-averaged photocatalytic activity for reductively prepared materials ($^{H,DP}M/TiO_2$ and $^{PD}M/TiO_2$, see Figure 2 and Table S1) is most likely related to the ubiquitous presence of reduced metallic nanoparticles, providing active surfaces for $H_2$ evolution. The case of copper might be slightly different, since oxidation to Cu(II) is known to take place in contact with atmospheric oxygen, regardless of the loading method. However, copper oxide species on $TiO_2$ produced by passivation under ambient air might be readily and reversibly reduced during irradiations and in the presence of glucose, acting as an electron donor. This behavior is commonly observed for similar Cu/$TiO_2$ systems [33,40–44]. A deeper study on the oxidation states and surface speciation of these photocatalysts will be described elsewhere, thus complementing the preliminary results presented herein.

Regarding the suitability of copper as a co-catalyst for photocatalytic $H_2$ evolution, it is noteworthy that activities on a par with noble metal contenders were achieved. For example, $H_2$ production rates were essentially identical for $^{H,DP}Cu/TiO_2$ and $^{H,DP}Au/TiO_2$ (872.6 and 879.2 $\mu$mol $g_{cat}^{-1}$ $h^{-1}$, respectively, see Figure 2 and Supporting Information). The activity of the Pt-loaded material is substantially higher (almost double, 1812.5 $\mu$mol $g_{cat}^{-1}$ $h^{-1}$), yet this is not surprising considering that platinum has been reported as the best metal co-catalyst in a large number of systematic studies on photoreforming [7]. However, given the significantly lower cost and higher availability of copper as compared to platinum and gold [45], the notable performance of $Cu/TiO_2$ materials makes them as serious contenders regarding overall process cost in the design of solar hydrogen production from biomass-derived substrates [46]. In this report, we will insist on the idea of using copper as a viable alternative to noble metals for the photocatalytic production of $H_2$ from biomass or wastewaters.

### 2.3. Photocatalytic Hydrogen Production form Biomass Hydrolysates

Once the photocatalytic hydrogen production activity was tested for the $M/TiO_2$ materials in a systematic fashion, the next step in our research strategy was to prove the potential applicability of the solar photoreforming technology to real biomass hydrolysates, as proposed in the Introduction. In the first place, $\alpha$-cellulose was considered as a model lignocellulosic biomass substrate, since it is not fully a purified cellulose product, but a mixture of cellulose and other polysaccharides (mostly xylan) readily obtained from raw wood or agricultural crops after treatment in strongly basic aqueous media [47]. Then, the direct use of rice husks was pursued as a further step in the approach to valorize biomass waste.

The hydrolysis of $\alpha$-cellulose was performed by a mild procedure based on a short acid-catalyzed treatment carried out by raising the temperature of an aqueous suspension of the substrate to a desired peak value, and subsequently allowing the mixture to cool down to room temperature (see Section 3.4). The avoidance of an isothermal hydrolysis step at relatively high temperatures (above 160 °C) is expected to minimize the degradation of monomeric saccharides in consecutive reactions. In fact, this procedure allowed optimized yields of alkyl glycoside surfactants in an analogous cellulose alcoholysis method developed by our team, especially when performed for several short thermal cycles [48].

Product yields, $\alpha$-cellulose conversion and molar balances are listed in Table S2 for hydrolysis experiments performed at two different temperatures, that is, 160 and 180 °C, revealing that the lower temperature leads to the production of monomeric saccharides (glucose and xylose) with no sign of further degradation—and thus, essentially quantitative molar balances—yet at the expense of moderate conversions. It is interesting to note that very high xylose yields (93.6%) were achieved after three hydrolysis cycles according to the theoretical amount of xylan in the starting $\alpha$-cellulose, whereas glucose yields were noticeably lower (16.2%). In contrast, hydrolysis cycles at 180 °C result in higher conversions, albeit degradation of saccharides in solution becomes significant (see Table S2). Based on these data, mild hydrolysis at 160 °C was chosen as the procedure to obtain biomass-derived aqueous streams for subsequent photocatalytic hydrogen production.

An $\alpha$-cellulose hydrolysate resultant from a first cycle at 160 °C as described above (and in Section 3.4) contained glucose and xylose in sufficient concentrations (0.53% and 0.85% $w/v$, respectively, see Table S2) for measurable and significant $H_2$ production rates to be observed. However, the results of the photocatalytic experiments resulted in poor hydrogen evolution performance (immeasurable yields after 2 h, Table S3) for $^{H,DP}Au/TiO_2$, $^{H,DP}Cu/TiO_2$, and even for the most active photocatalyst in the screening reported in Section 2.2 above, $^{PD}Pt/TiO_2$. Although $H_2$ production rates dropped by several orders of magnitude as compared to the pure glucose solutions (see Figure 2 and Table S1), the formation of $CO_2$ was relevant and of a similar magnitude (in the order of $10^2$ $\mu$mol $g_{cat}^{-1}$ $h^{-1}$) than for model systems. Long-term irradiations were then performed in order to check whether $H_2$ evolution would start after some sort of induction period, as for some related photocatalytic systems reported by the Kondarides' team [49–51]. No such phenomenon was observed (see Figure 3).

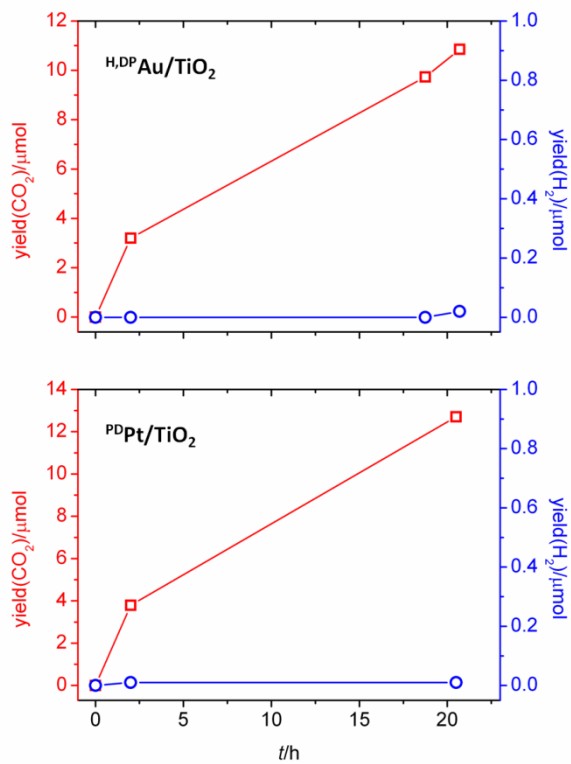

**Figure 3.** Long-term photocatalytic yields of $CO_2$ (red squares) and $H_2$ (blue circles) from the $\alpha$-cellulose hydrolysate obtained at 160 °C (cycle #1, see Section 3.4 and Table S2 footnote) on $^{H,DP}$Au/TiO$_2$ (top) and $^{PD}$Pt/TiO$_2$ (bottom). Reaction conditions: Stirred suspensions of the photocatalyst (25 mg) in $\alpha$-cellulose hydrolysate (25 mL) irradiated under simulated solar light (AM1.5G, 1.0 kW m$^{-2}$) under Ar atmosphere (1.4 bar) at 25 °C.

The reason behind the negligible $H_2$ production rates from $\alpha$-cellulose hydrolysates under simulated sunlight might be due to a series of factors. The acidity of the aqueous media, carrying 1% concentrated hydrochloric acid, might detrimentally affect the hydrogen evolution performance of the photocatalytic system, as suggested in a series of studies [7]. In order to check and ponder the magnitude of such an effect, an experiment using an acidified glucose solution at the same HCl concentration as the $\alpha$-cellulose hydrolysates was performed (see Figure S1). In an acidic medium, the photocatalytic $H_2$ production slows down to ca. 40% of the original rate. Nevertheless, this reduction is not as drastic as to account for the immeasurable $H_2$ yield from hydrolysates.

Another possible reason for the lack of $H_2$ evolution from $\alpha$-cellulose hydrolysates might lie in the generation of minor products, which inhibit such a photocatalytic reaction. This was proposed by Shende et al. for processed lignocellulosic biomass streams, and the hypothesis proven by removing colored (chiefly aromatic) impurity byproducts by adsorption onto active carbon [22,23]. After their decolorisation treatment, activity towards $H_2$ production was restored to some extent, albeit inevitable removal of valuable oxygenates by adsorption was also observed. Kondarides and co-workers demonstrated that aromatic dye molecules can in fact be photoreformed, but requiring extended irradiation times for the onset of $H_2$ evolution, especially at low pH and moderately high concentrations [50].

Herein, we propose a different intermediate treatment for lignocellulosic biomass hydrolysates aiming at no loss of organic matter by adsorption (or by any other means), and hence, keeping all of it in solution and susceptible to be converted into $H_2$. In this regard, the forced photoreforming of the presumably resistant colored impurities by energy-intensive UV irradiation is considered. Sunlight irradiation contains a low proportion of high-energy photons in the UV range (<400 nm), and thus, photocatalytic processes on TiO$_2$-based materials might be relatively slow, especially for highly

energy-demanding redox reactions [52]. This approach was tested for the $\alpha$-cellulose hydrolysate considered above on $^{\text{H,DP}}$Cu/TiO$_2$ (Table S3). The UV (Hg lamp) irradiation on the raw hydrolysate solution resulted in a noticeable production of H$_2$ (17.3 $\mu$mol g$_{\text{cat}}^{-1}$ h$^{-1}$), a clear improvement relative to the analogous process using simulated sunlight, as clearly shown in Figure 4. Direct photocatalytic evolution of hydrogen without any physicochemical pre-treatment of this processed cellulose aqueous stream reinforces the strategy advocated in this work about total biomass-to-hydrogen conversion based on hydrolysis and subsequent photocatalysis (see Introduction, Figure 1). After the initial UV treatment, the amber coloration of the aqueous $\alpha$-cellulose hydrolysate stream had disappeared largely, and a faint yellowish (almost colorless) solution was obtained (Figure S2), a fact that may be described as bleaching with concomitant H$_2$ evolution. Irradiation of this UV-treated hydrolysate under simulated sunlight resulted in a significantly improved H$_2$ production rate (more than 20-fold increase, Figure 4).

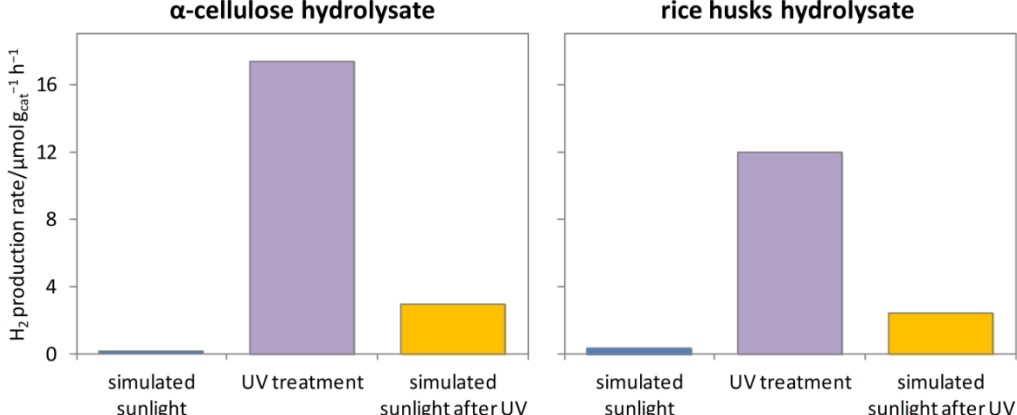

**Figure 4.** Photocatalytic H$_2$ production yields from $\alpha$-cellulose (cycle #1, see Section 3.4 and Table S2 footnote) or rice husks hydrolysates obtained at 160 °C (see Table S4) on $^{\text{H,DP}}$Cu/TiO$_2$ by direct simulated sunlight irradiation (AM1.5G, 1.0 kW m$^{-2}$), treatment under UV (Hg lamp, $\approx$ 1.5 kW m$^{-2}$) light, and simulated sunlight irradiation after this previous UV treatment. Reaction conditions: Stirred suspensions of the photocatalyst (25 mg) in the biomass hydrolysate (25 mL) irradiated under Ar atmosphere (1.4 bar) at 25 °C, *t* = 2 h.

Since the byproducts presumably causing inhibition of H$_2$ production are expected to be minor, but highly absorbing, their evolution was monitored by UV-vis absorption spectroscopy. As seen in Figure 5, the raw $\alpha$-cellulose hydrolysate contained species causing strong absorption below 350 nm and also extending well into the visible range, with a clear shoulder at ca. 580 nm. After UV irradiation, a noticeable decrease in absorption was observed at those frequencies, a fact that might reinforce the hypothesis of decolorisation as a strategy to boost photocatalytic H$_2$ production. However, careful comparison of the decolorizing effect under either Hg lamp or simulated solar light revealed little differences (see Figure 5, right, visible range). This indicates that removal of chromophores does not result in a significant improvement of the photocatalytic activity, since decolorisation is effective under simulated sunlight, but no significant H$_2$ evolution was observed (Figure 4, Table S3). Conversely, a closer look at higher frequencies hints noticeable differences (see Figure 5, left, UV range, 300–330 nm): UV irradiation causes a significant absorption decrease, whereas the effect is clearly less obvious under simulated sunlight. Typically, polyaromatic or heteroaromatic (including furanic) moieties absorb in this region [53,54]. Therefore, the reason for the apparent activation of the system after UV treatment might be the light-activated removal (via photoreforming) of small amounts either furanic or (less likely) polycyclic aromatic byproducts formed by degradation of saccharide species during the acidic hydrolysis step. These species are presumably inhibitory of the reductive H$_2$ evolution processes.

Whilst subsequent solar photoreforming did not result in further removal of these inhibiting species (Figure 5, left), decolorisation continued steadily (Figure 5, right).

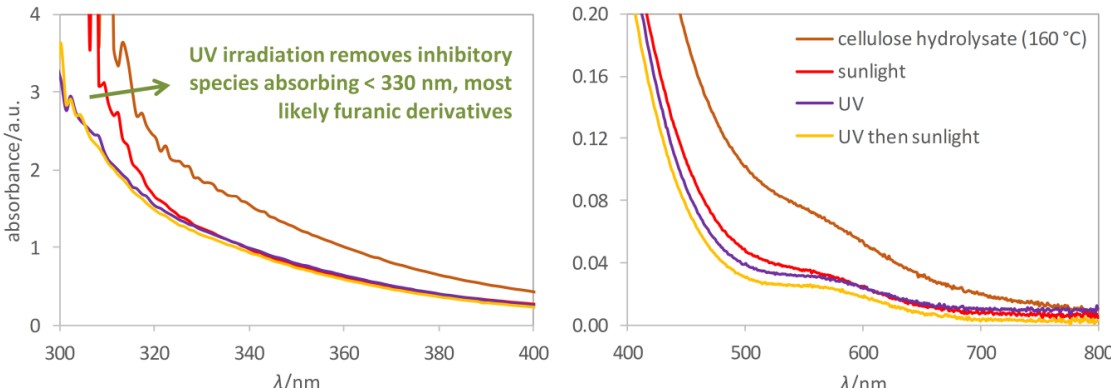

**Figure 5.** UV-vis absorption spectra of the raw α-cellulose hydrolysate obtained at 160 °C (cycle #1, see Section 3.4 and Table S2 footnote), and after irradiations under direct simulated sunlight, UV (Hg lamp) light, and simulated sunlight after this previous UV treatment (see Table S3 for details). The left and right graphs illustrate the effects observed in the UV and visible regions, respectively.

*2.4. Photocatalytic Hydrogen Production form Wastewaters*

Going one step forward in assessing the applicability of solar photoreforming as a waste valorization technology, two kinds of wastewaters were considered as the feedstock for $H_2$ production, that is (i) municipal wastewaters, and (ii) industrial wastewaters from a plant producing fruit juices. The former were chosen since they are produced in huge amounts in cities and towns, whereas the latter represent a more limited stream, yet containing higher amounts of oxygenated substances, chiefly saccharides, prone to undergo efficient photoreforming. Compositional characteristics of the two wastewater streams employed in this work are listed in Table S4.

Irradiation of $^{H,DP}M/TiO_2$ (M = Cu or Au) suspensions in samples of either municipal or juice production wastewaters were performed using simulated sunlight irradiation, and the results listed in Table S5 and plotted in Figure 6. The production of $H_2$ was almost negligible from municipal wastewaters ($\leq 0.1$ μmol $g_{cat}^{-1}$ $h^{-1}$) despite the relatively high content of organic matter (see Table S4). It is worth noting that the photocatalytic activity is not zero, as suggested by the noticeable evolution of $CO_2$ (Table S5), which shows that oxidation of organic species can still take place. A different outcome was observed for wastewaters sourced by a juice producing plant. Noticeably high $H_2$ productions were recorded using a juice production wastewater with no other treatment than avoiding some settled solids (Figure 6). The photoreforming process is especially efficient using $^{H,DP}Au/TiO_2$ as the photocatalyst, with production rates remarkably higher than for municipal wastewaters (by three orders of magnitude, 115 μmol $g_{cat}^{-1}$ $h^{-1}$, see Table S5). This is surprising since, based on HPLC data, the estimated chemical oxygen demand (COD) for the fruit production wastewater sample is comparable (several hundred mg($O_2$) $L^{-1}$, see Table S4) to that of the municipal treatment plant. Copper as a co-catalyst performed at lower yields, although this system might be further optimized by ensuring extensive copper reduction in situ, as shown for similar photocatalytic systems [33]. Concomitantly, $CO_2$ productions were even faster for both materials (Table S5).

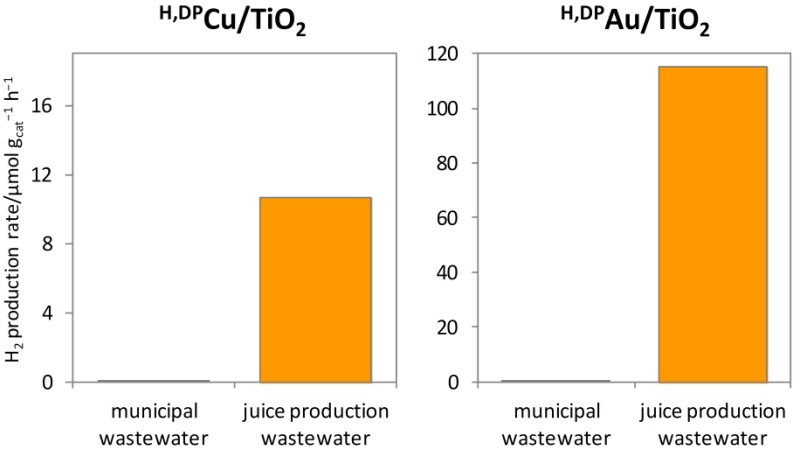

**Figure 6.** Photocatalytic $H_2$ production yields from either municipal or juice production wastewaters on $^{H,DP}Cu/TiO_2$ or $^{H,DP}Au/TiO_2$. Reaction conditions: Stirred suspensions of the photocatalyst (25 mg) in the corresponding wastewater sample (25 mL) under simulated sunlight irradiation (AM1.5G, 1.0 kW m$^{-2}$) under Ar atmosphere (1.4 bar) at 25 °C.

The photocatalytic $H_2$ production from municipal wastewaters has been previously reported by Malato and co-workers as a process of limited efficiency using $M/TiO_2$ ($M$ = Pt [55], Au [56] or Cu [57]). The authors attributed the low activity to the presence of different kinds of solutes inhibiting $H_2$ evolution, in particular ionic species, albeit the detrimental effect of recalcitrant organic matter cannot be ruled out. In one of these reports, the use of wastewater samples from an orange juice bottling plant was also used as the feedstock, and contrarily to the data reported herein, $H_2$ production rates were lower than for samples of municipal wastewaters [56]. As in our case, no direct relationship was found between the concentration of organic matter and photoreforming activity, and instead, it was proven that the highly saline nature of the aqueous stream hindered $H_2$ production. In order to check whether ionic strength might be also a decisive factor for our samples, conductivities were measured, and as opposed to Malato's work mentioned above, salinity was not detrimental for the juice production wastewater studied herein (exhibiting a > 30% higher conductivity as compared to municipal wastewater sample, see Table S4) to become a suitable substrate for photocatalytic hydrogen production. This reflects the importance of multiple factors strongly affecting photoreforming activity for the valorization of wastewaters. Similar samples might give rise to greatly diverging efficiencies depending on the compositional characteristics and history of each individual wastewater stream.

## 3. Materials and Methods

### 3.1. Materials

Titanium dioxide (Aeroxide® $TiO_2$ P25) was sourced by Evonik. Hydrogen hexachloroplatinate(IV) hexahydrate (≥99.9% free of trace metals), chloroplatinic acid solution (8% *w* in $H_2O$), hydrogen tetrachloroaurate(III) trihydrate (≥99.9% free of trace metals), silver nitrate (>99%), copper(II) nitrate hemi(pentahydrate) (98%), copper(II) nitrate trihydrate (>99%), hydrochloric acid (37%), D-(+)-glucose (ACS reagent), and α-cellulose (powder, estimated composition of ideal hydrolysate ≈ 85:15 *w/w* glucose/xylose) were supplied by Sigma-Aldrich and used as received. Methanol (reagent grade, ≥99.9%) was supplied by Scharlau and used as received. Sodium hydroxide (pellets ≥ 99%) was supplied by Merck and used as received. Ultra-pure water was obtained using a Milli-Q® purification system. Argon (≥99.995%) were supplied by Abelló Linde. The municipal wastewater sample used in this work was obtained from the influent stream to the Carraixet wastewater treatment plant (Alboraia, Spain) on the 27th of November 2018; its chemical oxygen demand (COD) was 389 mg L$^{-1}$, and its solids content was 316 mg L$^{-1}$, as reported by the plant analysis laboratories. Solids were removed by filtration through standard filter paper prior to photocatalytic experiments. The wastewaters derived

from fruit juice production operations were generously sourced from a plant operated by the AMC Natural Drinks S.L. (Murcia, Spain). Rice husks were kindly supplied by a local rice mill (Arroces Lozano, Arroz el Cazador, Alginet, Spain) and milled to a powder (<1 mm) using a coffee grinder.

### 3.2. Photocatalyst Syntheses

The $M$/TiO$_2$ ($M$ = Pt, Au, Ag or Cu; aiming at a 1% *w/w* $M$/TiO$_2$ loading) photocatalysts used in this work were prepared by either deposition-precipitation followed by hydrogenation ($^{H,DP}M$/TiO$_2$), photodeposition ($^{PD}M$/TiO$_2$) or impregnation followed by calcination ($^{IM}M$/TiO$_2$) methods, as listed in Table 2.

In a typical deposition-precipitation synthesis, TiO$_2$ (1.00 g) was dispersed in a certain volume of ultra-pure water containing the desired amounts of metallic precursor, as detailed in Table 2. The suspensions were vigorously stirred while their pH was adjusted to the desired values (see Table 2), using aqueous NaOH (0.2 M) and a pH-meter (SevenMulti Metler Toledo). The stirring was maintained at room temperature during the specified time while bubbling N$_2$ through the suspension in order to avoid atmospheric CO$_2$ absorption. The solids were then separated by filtration using cellulose acetate membranes (Whatman®, pore size = 0.2 μm), and washed with deionized water until the complete elimination of chloride ions (where appropriate), as checked by the AgNO$_3$ test. The solids were vacuum dried (≈10 mbar) at room temperature until constant weight was achieved and subsequently milled to a fine powder with a mortar and pestle. The powder was finally treated under H$_2$ flow (50 mL min$^{-1}$) at 300 °C (heating ramp at 3 °C min$^{-1}$) for 5 h to obtain the final $^{H,DP}M$/TiO$_2$ photocatalysts.

In a typical photodeposition procedure, TiO$_2$ (200 mg) was dispersed in ultra-pure water (25 mL) containing the desired amounts of metallic precursor and methanol (80 mg), as detailed in Table 2. The suspensions were sonicated for 10 min and then transferred to cylindrical quartz cells (diameter ≈ 44 mm, volume ≈ 50 mL, equipped with a gas inlet valve, a gas outlet valve and a pressure gauge). The cells were evacuated under vacuum (≈10 mbar, 2 min) and then purged with argon (pressurizing up to 2 bar and depressurizing, five cycles). The cells were loaded with argon (1.5 bar) and then the stirred (500 min$^{-1}$) suspensions irradiated under UV-vis light from a mercury lamp (125 W, water-refrigerated, irradiance ≈ 1.5 kW m$^{-2}$, measured employing a calibrated photodiode) for 3 h. The suspensions turned from off-white to intensely colored at the end of the photodeposition processes. The solids were separated by filtration using polyamide membrane filters (Whatman®, pore size = 0.45 μm), washed with ultra-pure water (ca. 0.3 L). The resulting pastes were further dried under vacuum (≈10 mbar) at room temperature until constant weight and then ground to a powder using a mortar and pestle to obtain the final $^{PD}M$/TiO$_2$ photocatalysts.

In a typical impregnation procedure, the required amount of metallic precursor was dissolved in ultra-pure water (1.2 mL). The solution was then evenly dropped on TiO$_2$ (1.00 g) and the mixture manually stirred with a plastic spatula until homogenous. The resulting paste was subsequently vacuum dried (≈10 mbar) at room temperature until constant weight, and the solid obtained was finally milled to fine powder with a mortar and pestle and calcined at 400 °C (heating ramp at 2 °C min$^{-1}$) for 3 h under static air to obtain the final $^{IM}M$/TiO$_2$ photocatalysts.

### 3.3. Characterisation and Analysis

Metallic contents were determined by ICP analyses performed on a Varian 715-ES ICP Optical Emission Spectrometer after dissolution of the samples in aqueous HF/HNO3/HCl (1:1:3) mixtures. In the case of silver (and also $^{IM}$Pt/TiO$_2$), ICP did not provide satisfactory results due to the formation of precipitates or deposits during dissolution, and therefore, elemental analyses were performed by means of FESEM-EDX on a JEOL 7001F microscope equipped with an Oxford Instruments detector, after depositing a layer of powdered sample on carbon tape; elemental percentages were averaged over a number of measurements on different areas of the specimen. Transmission electron microscopy (TEM) images were taken on JEOL 2100F microscope operating at 200 kV both in high-resolution

transmission (HRTEM) and STEM modes, coupled with an Inca Energy TEM 200 (Oxford) energy dispersive X-ray (EDX) spectroscope for elemental analyses. STEM images were obtained using a high angle annular dark field (HAADF) detector, which allows Z-contrast imaging. Samples were deposited on carbon-coated nickel grids.

Analyses of aqueous samples were performed after filtration using polyamide syringe filters on an Agilent Infinity HPLC instrument using an Aminex® HPX-87H column (injection volume = 5 μL, column temperature = 60 °C, eluent: 4 mM aqueous $H_2SO_4$, flow rate = 0.7 mL min$^{-1}$) and a refractive index detector. The UV-vis absorption spectra of α-cellulose hydrolysates before and after irradiations were recorded on a Varian Cary 50 Conc UV-visible spectrophotometer in 1 cm optical path length quartz cells. Conductivity of wastewaters was measured using a Crison CM 35 device equipped with a Crison + Pt 1000 Conductivity Cell.

### 3.4. Cellulose (Or Rice Husks) Hydrolysis Experiments

In a typical experiment, α-cellulose or rice husks powder (6.00 g, 37.0 mmol) were suspended in aqueous HCl (1:99 *v/v* mixture of concentrated HCl and water, *c*(HCl) ≈ 0.43% *w/v*, pH ≈ 0.9, 60 mL) inside a stainless steel autoclave equipped with a gas inlet valve, a rupture disk, a pressure gauge and a thermocouple, purged with $N_2$ (pressurizing to 20 bar and depressurizing to ambient pressure, three cycles), and tightly closed using a teflon seal. The suspension was stirred (1000 min$^{-1}$) and heated up to the desired temperature by means of a heating tape wrapped around the vessel, and immediately allowed to cool down to room temperature. Conversions were calculated gravimetrically after separating the unreacted solid by filtration through a conventional filter paper, washing with deionized water (5 × 10 mL), and drying the remaining solid under reduced pressure (ca. 10 mbar) until constant weight. The resulting unreacted solid was treated in subsequent hydrolysis steps up to a total of three cycles. Analyses of liquid samples were performed by HPLC as described in Section 3.3 above.

### 3.5. Photocatalytic Reactions

In a typical experiment, the photocatalyst powder (25 mg) was suspended in the desired aqueous reaction medium (either glucose solutions, biomass hydrolysates or wastewaters, 25 mL) by sonication for 10 min. The resulting suspension was then transferred to a cylindrical quartz reactor analogous to the one used for photodepositions (see Section 3.2). The cell was evacuated under vacuum (≈10 mbar, 2 min) and then purged with argon (pressurizing up to 2 bar and depressurizing for five cycles), and finally loaded with argon (1.5 bar). The suspension was stirred (500 min$^{-1}$) and irradiated using a solar simulator (ThermoOriel 91192-1000, equipped with a 1000 W Xe lamp and an AM1.5G filter to simulate the spectrum of sunlight; irradiance ≈ 1.0 kW m$^{-2}$, as measured using a calibrated photodiode). Gas phase samples (2 cm$^3$) were taken and analyzed on a two-channel chromatograph (Agilent 490 Micro GC, carrier gas: Ar) equipped with thermal conductivity detectors (TCD), a MS 5 Å column (first channel) for the quantification of $H_2$ and a PoraPLOT Q column (second channel) for the quantification of $CO_2$ and $C_{1-3}$ hydrocarbons. Estimated standard deviations for photocatalytic production rates lie within ± 5% error bars, as determined by independent results. Analyses of liquid samples were performed by HPLC as described in Section 3.3 above.

For photocatalytic UV-treatment experiments, the procedure was essentially identical to that described above for sunlight irradiations, but using a Hg lamp (125 W, irradiance ≈ 1.5 kW m$^{-2}$).

## 4. Conclusions

Titania-based materials with loaded metallic co-catalyst nanoparticles are well-known to perform efficiently for a range of photocatalytic reactions. Among these, reductive production of hydrogen coupled with the oxidation of organic substances, i.e., photoreforming, is of relevance given that high $H_2$ production rates can be achieved, representing an energetic valorization option for waste biomass feedstocks or even wastewaters rich in organic matter. The photocatalytic activity for a given process depends on a number of factors, especially on the type of metal co-catalyst and deposition method.

The systematic study performed in this work using glucose as a model substrate for more complex lignocellulose hydrolysates or food industry wastewaters revealed that deposition-precipitation (followed by reduction under hydrogen) or photodeposition resulted in improved activity over classic impregnation methods. Moreover, co-catalyst efficiencies decreased in the Pt > Au > Cu > Ag order, though differences between Cu and noble counterparts were small in some cases, thus inferring that copper represents a sustainable and inexpensive alternative component of $H_2$-producing photocatalysts.

Direct short-term application of solar photocatalysis to mildly hydrolyzed lignocellulosic materials ($\alpha$-cellulose or rice husks) resulted in negligible $H_2$ production, although noticeable $CO_2$ evolution and conversion of at least a portion of organic matter were observed by UV-vis spectroscopy. Thus, activity was not suppressed totally, but instead, inhibition of the reductive hydrogen production caused by minor amounts of recalcitrant hydrolysis byproducts occurred. A novel method to break down these inhibitors based on a photocatalytic UV treatment is herein proposed and proven. This allows $H_2$ production to start immediately (at 17 $\mu$mol $g_{cat}^{-1}$ $h^{-1}$ on $Cu/TiO_2$) without any intermediate physicochemical treatment such as adsorption, which is likely to cause loss of valuable biomass-derived organic matter. Finally, it is shown that photoreforming of organic matter in municipal wastewaters is similarly inhibited, in contrast to certain types of aqueous food industry streams, as samples of juice production wastewaters containing significant levels of saccharides ($H_2$ yields amounted to 115 $\mu$mol($H_2$) $g_{cat}^{-1}$ $h^{-1}$ using $Au/TiO_2$). For the particular samples tested herein, high $H_2$ production rates from juice production wastewaters make the application of photoreforming for energetic valorization of certain types of industrial waste possible and promising.

**Supplementary Materials:** The following are available online at http://www.mdpi.com/2073-4344/9/7/584/s1: Figure S1. Photocatalytic $H_2$ yields from aqueous glucose at either natural pH or in an HCl-acidified medium on $^{H,DP}Au/TiO_2$. Figure S2. Picture showing the amber colour of the $\alpha$-cellulose hydrolysate obtained at 160 °C, and the bleaching effect caused by the UV photocatalytic treatment leading to an almost colourless solution. Table S1. Product yields for the photocatalytic reforming of glucose under simulated solar light on $M/TiO_2$. Table S2. Hydrolysis of $\alpha$-cellulose in acidic aqueous media in short thermal cycles at different temperatures. Table S3. Product yields for the photocatalytic reforming of the $\alpha$-cellulose hydrolysate obtained after a first hydrolysis cycle at 160 °C under simulated solar light either directly or after UV pre-treatment on $M/TiO_2$. Table S4. Chemical characteristics of the rice husks hydrolysate, and wastewaters used in photocatalytic experiments. Table S5. Product yields for the photocatalytic reforming of wastewaters under simulated solar light on $M/TiO_2$.

**Author Contributions:** Investigation, M.I. and A.V.P.; Project administration, A.V.P.; Supervision, A.V.P.; Writing—original draft, M.I. and A.V.P.; Writing—review & editing, A.V.P.

**Funding:** Financial support from the Spanish Government (Ministry of Science, Innovation and Universities) through its "Severo Ochoa" excellence program (SEV 2016-0683), and to the Spanish Government (Agencia Estatal de Investigación) and the European Union (European Regional Development Fund) via a grant for young researchers (CTQ2015-74138-JIN, AEI/FEDER/UE), is acknowledged.

**Acknowledgments:** A.V.P. thanks the Spanish Government (Agencia Estatal de Investigación) and the European Union (European Regional Development Fund) for a grant for young researchers (CTQ2015-74138-JIN, AEI/FEDER/UE). M.I. thanks the Spanish Catalysis Society (SECAT) for a Masters Grant. AMC Innova S.L. (as part of AMC Natural Drinks S.L., Murcia, Spain) is gratefully acknowledged for providing samples of their process waters. Arroces Lozano and Arroz el Cazador (Alginet, Spain) are gratefully acknowledged for generously providing a rice husks sample. Carraixet wastewater treatment plant (Alboraia, Spain) is gratefully acknowledged for kindly providing samples of their influent streams. The authors also thank the Microscopy Service of UPV for kind help on TEM measurements.

**Conflicts of Interest:** The authors declare no conflict of interest. The funders had no role in the design of the study; in the collection, analyses, or interpretation of data; in the writing of the manuscript, or in the decision to publish the results.

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
