# Peer review of "Assessment of Photocatalytic Hydrogen Production from Biomass or Wastewaters Depending on the Metal Co-Catalyst and Its Deposition Method on TiO2"

_catalysts, doi:10.3390/catal9070584_

Round 1

Reviewer 1 Report

It is a well-written paper on photocatalytic production of hydrogen on M/TiO2 systems. It is particularly interesting since it explores the possibility to use more “real” samples (e.g. wastewaters) instead of lab mixtures and the way to primary degrade those byproducts present in the starting solutions probably responsible for the poor catalytic performance.

Nevertheless, to the referee´s mind some major points should be covered before it can be recommended for publication.

1)     Experimental part. It is understandable that the authors had to resort to other technique (EDX) instead of ICP for determination of Ag content in the samples. However, error values should be also included for contents determined by ICP. Otherwise, it sounds strange to the reader. By the way, what about the Ag and Pt contents in IMAg/TiO2 and IMPt/TiO2 samples, respectively? It is also difficult to understand that the experimental loading is higher than the nominal in some cases. If it is a result of hydration of starting materials, the experimental metal loading would be lower. In any case the water present in the precursor formula, used for calculations, is constitutional water (not hydration water). Maybe the error values could be of help to interpret the results.

2)     In order to make the authors´ results comparable to some others, the metal particle size should be determined (at least in the most active solids). In fact, the results expressed as micromole H2 per gram of catalyst and hour could change if expressed as TOF once the metal particle size has been determined.

3)     Determination of metal oxidation state by XPS (before and after the reaction, especially for the selected supported copper catalyst) would be also of help to see if there is in-situ reduction of metals as the reaction proceeds.

4)     Why DPCu/TiO2 instead of most active PDCu/TiO2 (Figure 2) is selected for further studies?

5)     The authors comment at one point on the possible detrimental effect of salinity of wastewaters on catalytic production. Maybe if would be useful to describe the starting water compositions not only in terms of COD but also include other parameters such as conductivity.

Once these points have been covered the manuscript would be definitely a reference for photoreforming studies using “real” biomass-derived and wastewater samples.

Author Response

It is a well-written paper on photocatalytic production of hydrogen on M/TiO2 systems. It is particularly interesting since it explores the possibility to use more “real” samples (e.g. wastewaters) instead of lab mixtures and the way to primary degrade those byproducts present in the starting solutions probably responsible for the poor catalytic performance.

We are sincerely grateful for the positive comments from the reviewer.

Nevertheless, to the referee´s mind some major points should be covered before it can be recommended for publication.

1)     Experimental part. It is understandable that the authors had to resort to other technique (EDX) instead of ICP for determination of Ag content in the samples. However, error values should be also included for contents determined by ICP. Otherwise, it sounds strange to the reader. By the way, what about the Ag and Pt contents in IMAg/TiO2 and IMPt/TiO2 samples, respectively? It is also difficult to understand that the experimental loading is higher than the nominal in some cases. If it is a result of hydration of starting materials, the experimental metal loading would be lower. In any case the water present in the precursor formula, used for calculations, is constitutional water (not hydration water). Maybe the error values could be of help to interpret the results.

The issue about metallic content in the photocatalysts raised by the reviewer is indeed of utmost importance and needed to be dealt with. Efforts have been made to complete ICP results with missing data and error bars. The metal contents for both IMAg/TiO2 and IMPt/TiO2 has been obtained by FESEM-EDX, and the data included in Table 1 (in addition, a short note has been included in the Materials and Methods Section 3.3 to state that also the Pt material had to be analysed by this method). In some cases, standard deviations were calculated from a minimum of two ICP measurements, whereas in others, an estimation was made based on instrument calibration and data from other similar samples (see Table 1 and its footnote). The errors from ICP are, as expected, much smaller than from FESEM-EDX, and thus, deviations from the nominal values are most likely real and due to an unknown amount of both constitutional and hydration water in the commercial metallic precursors, as confirmed as an example for HAuCl4.3H2O, for which a stock solution was found to contain 10% more gold than expected. This fact has been commented in detail in the text (Section 2.1, page 3).

2)     In order to make the authors´ results comparable to some others, the metal particle size should be determined (at least in the most active solids). In fact, the results expressed as micromole H2 per gram of catalyst and hour could change if expressed as TOF once the metal particle size has been determined.

The reviewer has made an important point here regarding co-catalyst morphology, in particular particle size as a key parameter affecting photocatalytic activity. We aim at extending the characterisation on these materials into a systematic and thorough study to be published elsewhere in the near future. As part of such study, analysis of particle sizes and other morphological aspects, and their relationship with performance was underway. However, we agree with the reviewer that particle sizes are indeed relevant to the present manuscript, and have thus incorporated data taken from STEM images into the main text (please see Table 1). Furthermore, a detailed description on the effect of co-catalyst loading method on particle sizes is now included in Section 2.1 (pages 3-4). The recommendation of the reviewer to calculate TOFs has also been followed (results are included in Table S1), revealing interesting facts, as for example the increase in site-specific activity with particle sizes, and some hints on the effect of deposition method on photocatalytic activity. Please see additions made to Section 2.2 (pages 5-6) commenting on this topic. In the interim of writing our complete account on the characterisation of the materials, which will provide further insight into the relationship between composition or morphology with activity, these data are clearly useful here.

3)     Determination of metal oxidation state by XPS (before and after the reaction, especially for the selected supported copper catalyst) would be also of help to see if there is in-situ reduction of metals as the reaction proceeds.

We totally agree with the reviewer that the determination of oxidation states and speciation of co-catalysts is crucial to find answers to the questions that the trends in activity might raise. As mentioned above, our intention is to perform XPS measurements on all materials, and ideally follow their evolution upon use under irradiation in the studied processes. Work is underway to obtain the required data, but due to time constrains we are not able to include this important part of the project in this manuscript, and limit its scope to the applicability of the photocatalysts to real biomass-derived streams. We therefore regret to keep most of the characterisation for our next publication on these systematic studies.

4)     Why DPCu/TiO2 instead of most active PDCu/TiO2 (Figure 2) is selected for further studies?

The reviewer is right in pointing out that the most active Cu material should be further studied, and that should be the case for a comprehensive optimisation work. DPCu/TiO2 was used herein for further experiments since the amount of material per batch was much higher, and hence, a more practical approach could be followed using it. Time constrains prevent doing this here, but for future studies, we will consider the optimisation of photocatalysts prepared by both methods, with special emphasis on the photodeposition contender.

5)     The authors comment at one point on the possible detrimental effect of salinity of wastewaters on catalytic production. Maybe if would be useful to describe the starting water compositions not only in terms of COD but also include other parameters such as conductivity.

We gratefully acknowledge the reviewer to raise this important aspect. Conductivity measurements have been performed, and the results included in the Supplementary Information (see Table S4), and commented in Section 2.4 (pages 10-11). In our case, salinity did not seem to be a critically detrimental factor affecting activity, since the most suitable wastewater in terms of hydrogen production (the sample from industrial juice production processes) had a higher ionic content, based on a significantly higher conductivity (+ > 30% as compared to the municipal wastewater sample.

Once these points have been covered the manuscript would be definitely a reference for photoreforming studies using “real” biomass-derived and wastewater samples.

Reviewer 2 Report

- The article entitled “Assessment of photocatalytic hydrogen production from biomass or wastewaters depending on the metal co-catalyst and its deposition method on TiO2” presents outstanding results with a nice discussion and deserves publication in Catalyst MDPI. I, however, recommend authors follows some recommendations for R1 version.

- Please, include error bars in the catalytic figures (eg. Figure 2). 

- Photo-handling schemes could be useful as a summary of results discussions. Are there any differences in the expected charge schemes using different co-catalysts?

- Analysis of chemical environment seems to be very important in this contribution. Oxidation state, as well as the concentration of the co-catalysts on the surface, must be reported and discussed. 

- Please include more works as relevant references (See eg. Molecular Catalysis 437 (2017) 1–10 Contents, Applied Catalysis B: Environmental 238 (2018) 434–443 and other contributions from Prof. M. Fernandez-Garcia and Prof. E. Selli, among others expert in H2 photo-production reactions.  

Author Response

-           The article entitled “Assessment of photocatalytic hydrogen production from biomass or wastewaters depending on the metal co-catalyst and its deposition method on TiO2” presents outstanding results with a nice discussion and deserves publication in Catalyst MDPI. I, however, recommend authors follows some recommendations for R1 version.

We are sincerely grateful for the positive comments from the reviewer.

-           Please, include error bars in the catalytic figures (eg. Figure 2). 

Following on the suggestion by the reviewer, we have made an effort to quantify error bars for photocatalytic performance parameters based on a number of independent replicated experiments in the framework of our “Photocatalytic production of hydrogen from biomass derivatives” project. This has revealed that the error bars generally lie within a ± 5% range. This is now stated in the caption of Figure 2 and is intended to be an estimation of the actual standard deviations for each individual reaction, which would be exceedingly time- and resource-consuming. A mention to this has been also included in the Materials and Methods Section (3.5, page 14).

-           Photo-handling schemes could be useful as a summary of results discussions. Are there any differences in the expected charge schemes using different co-catalysts?

Our work systematically compares different metal co-catalysts deposited on the same support (TiO2, P25). Therefore, energy levels are expected to be similar, except for the position of the Fermi level, which might be dictated by the work function of the metal. Since work functions can vary depending on surface morphology even for the same metal, the design of energy level schemes with a sufficiently high confidence on the position of each level on the energy scale is slightly ambitious at this point for us given the limited amount of time provided for revision of this work. We take the reviewer suggestion as matter for future discussion and consideration. A theoretical approach such as that hinted by the reviewer might be included in our next manuscript dealing with morphology-activity relationships of the title materials.

-           Analysis of chemical environment seems to be very important in this contribution. Oxidation state, as well as the concentration of the co-catalysts on the surface, must be reported and discussed. 

Indeed, it is our aim to elucidate the chemical speciation (oxidation states) and physical characteristics of the metal deposits, and to this end, XPS, DRUV, and other spectroscopic investigations are underway. Regrettably, we are not able to include such data in this manuscript due to time limitations, but instead, a great deal of characterisation studies will be included in our next publication on the materials studied herein. We thank the reviewer to point out that not only oxidation states, but also concentration and distribution are important, and as a first approximation to this, average and standard deviations for particle sizes are reported in Table 1.

-           Please include more works as relevant references (See eg. Molecular Catalysis 437 (2017) 1–10 Contents, Applied Catalysis B: Environmental 238 (2018) 434–443 and other contributions from Prof. M. Fernandez-Garcia and Prof. E. Selli, among others expert in H2 photo-production reactions.  

We thank the reviewer for suggesting such insightful references, which have been checked carefully and cited as competently performed works on morphology-activity relationships in similar photocatalytic systems (see page 5, bottom). The explicitly mentioned (and other) articles are now included in the references list:

36.       Caudillo-Flores U, Muñoz-Batista MJ, Cortés JA, Fernández-García M, Kubacka A (2017) UV and visible light driven H2 photo-production using Nb-doped TiO2: Comparing Pt and Pd co-catalysts. Molecular Catalysis 437:1-10

37.       Ouyang W, Muñoz-Batista MJ, Kubacka A, Luque R, Fernández-García M (2018) Enhancing photocatalytic performance of TiO2 in H2 evolution via Ru co-catalyst deposition. Applied Catalysis B: Environmental 238:434-443

38.       Fontelles-Carceller O, Muñoz-Batista MJ, Rodríguez-Castellón E, Conesa JC, Fernández-García M, Kubacka A (2017) Measuring and interpreting quantum efficiency for hydrogen photo-production using Pt-titania catalysts. Journal of Catalysis 347:157-169

39.       Naldoni A, D'Arienzo M, Altomare M, Marelli M, Scotti R, Morazzoni F, Selli E, Dal Santo V (2013) Pt and Au/TiO2 photocatalysts for methanol reforming: Role of metal nanoparticles in tuning charge trapping properties and photoefficiency. Appl. Catal., B 130-131:239-248

Reviewer 3 Report

There are some issues that should be addressed prior to publication. The abstract and conclusions sections should quantitatively summarize the findings of the work. Figure 1 needs a scale. Table 1, define the - entries. Figure 2, give error bars on all data pints. L194, define high. L213, take out it is worthwhile to note that. Figure 3, define the - entries. Figure 4, give error bars. Figure  , give error bars. Table 2, define the - entries. 

Author Response

We sincerely thank the reviewer for the positive marks regarding our manuscript.

Solid H2 yield data have been included in both the Abstract and Conclusions sections (see highlighted changes) to better illustrate the achievements made in photocatalytic valorisation of wastewaters and biomass hydrolysates.

The meanings of all blank entries in Tables 1 and 2 have been explained by using appropriate footnotes. In this way, ambiguity has been minimised in the presented data.

The term “high” regarding hydrolysis temperature has been defined in the text, according to the reviewer’s remark, as above 160 °C (see L195).

The superfluous beginning “it is worthwhile to note…” has been deleted from L215.

Figure 3 has been modified following the recommendations made by the reviewer, and now includes the codes of the photocatalysts corresponding to both plots.

Error bars for photocatalytic performance parameters have been estimated based on a number of independent replicated experiments, as stated in the caption of Figure 2 or in Section 3.5 (lines 444-445). Unfortunately, standard deviations for each particular data point cannot be determined within the limited time frame given for this revision, neither in a reasonable amount of time, given the large number of experiments involved.

The scheme shown in Fig. 1 is meant to provide a visual illustration on the projects presented in this manuscript, yet it is only a qualitative and conceptual flowchart. For this reason, we have not been able to include a scale to it, to the best of our reasoning.

Again, we appreciate the input by the reviewer and hope that our amendments have addressed his/her concerns.

Round 2

Reviewer 1 Report

The authors have covered the main points I had raised. Therefore, I recommend its acceptance.

Author Response

The reviewer is gratefully acknowledged for a positive and helpful input, helping the manuscript quality to improve.